# Effect of Drying Kinetics, Volatile Components, Flavor Changes and Final Quality Attributes of *Moslae herba* during the Hot Air Thin-Layer Drying Process

**DOI:** 10.3390/molecules28093898

**Published:** 2023-05-05

**Authors:** Min Xie, Ying Chen, Yong Sun, Yarou Gao, Zhenfeng Wu, Ruiyu Wu, Rui Li, Shixi Hong, Minyan Wang, Yiping Zou, Hua Zhang, Yaokun Xiong

**Affiliations:** 1Department of Pharmaceutics, College of Pharmacy, Jiangxi University of Chinese Medicine, Nanchang 330004, China; 2State Key Laboratory of Food Science and Technology, Nanchang University, Nanchang 330047, China; 3Department of Food Science, University of Guelph, Guelph, ON N1G 2W1, Canada

**Keywords:** *Moslae herba*, drying kinetics, volatile components, flavor change, bioactive compounds

## Abstract

*Moslae herba* is considered to be a functional food ingredient or nutraceutical due to its rich bioactive components. The present research was carried out to investigate the effects of different temperatures (40 °C, 50 °C and 60 °C) on the drying characteristics, textural properties, bioactive compounds, flavor changes and final quality attributes of *Moslae herba* during the hot air-drying process. The results showed that the Midilli model could effectively simulate the drying process of *Moslae herba*. The effective moisture diffusivity ranged from 3.14 × 10^−5^ m^2^/s to 7.39 × 10^−5^ m^2^/s, and the activation energy was estimated to be 37.29 kJ/mol. Additionally, scanning electron microscopy (SEM) images of *Moslae herba* samples showed the shrinkage of the underlying epidermal layers and glandular trichomes. In total, 23 volatile compounds were detected in *Moslae herba*. Among them, the content of thymol increased from 28.29% in fresh samples to 56.75%, 55.86% and 55.62% in samples dried at temperatures of 40 °C, 50 °C and 60 °C, respectively, while the other two components, p-cymene and γ-terpinene, decreased with an increase in the temperature. Furthermore, both radar fingerprinting and principal component analysis (PCA) of the electronic nose (E-nose) showed that the flavor substances significantly altered during the drying process. Eventually, drying *Moslae herba* at 60 °C positively affected the retention of total phenolics, total flavonoids and the antioxidant capacity as compared with drying at 40 °C and 50 °C. The overall results elucidated that drying *Moslae herba* at the temperature of 60 °C efficiently enhanced the final quality by significantly reducing the drying time and maintaining the bioactive compounds.

## 1. Introduction

*Moslae herba* is well known as a highly valuable plant and has been widely used in the flavoring and pharmaceutical industries [1]. It is a member of the *Labiatae* family and is mainly found in the southern regions of China, Vietnam, India and Japan [2]. Generally, *Moslae herba* is consumed in dry form and is recognized as a good source of bioactive substances with therapeutic potential, including essential oils, flavonoids, steroids and triterpenes [3,4]. The whole *Moslae herba* has a therapeutic effect on modulating headaches, diarrhea, edema, coagulation, stomach pain, dysphonia, nephritis and throat infections due to its nutritional and functional properties [5]. In recent years, *Moslae herba* has gained increasing interest from both consumers and researchers owing to its antioxidant, antibacterial, anti-inflammatory, antitumor and immunomodulatory attributes [6,7]. However, fresh *Moslae herba* is difficult to store due to its high moisture content and heat sensitivity, which may lead to low quality and a short shelf life. Therefore, it is necessary to take appropriate steps solve this problem to improve the quality of *Moslae herba* products.

Drying is one of the most effective approaches to preserving food by reducing the moisture content and inhibiting the growth and reproduction of microorganisms to extend the shelf life and ensuring the quality and stability of food products [8]. The quality of the properties of the dried food products are greatly dependent on the drying conditions and drying methods [9]. However, unfavorable drying conditions may result in the loss of their characteristic flavor substances and nutrients, as well as undesired structural changes, thus reducing the overall quality of food products and lowering their market value [10]. Traditional drying approaches such as air-drying in the shade are commonly applied to dry *Moslae herba*. Being a natural drying method of exposing products to sun and fresh air, the traditional drying process is economic but usually requires a longer drying time because of the absence of thermal resources. This makes the products susceptible to secondary infections and a loss of nutrients [11]. Hot air-drying processes have been widely applied in the food industry due to the advantages of high efficiency, less cost and great simplicity. The temperature during the whole drying process of hot air-drying can be well controlled, and drying equipment shows high efficiency compared with traditional drying [12,13]. In recent years, the effect of hot air-drying on various food products has been studied. For example, previous investigations have shown that the appearance and drying time of persimmon slices, and the physicochemical properties and, bioactive components and microstructure of mango slices were significantly changed during hot air-drying [13,14].

The quality and nutritional value of food are required to satisfy the needs of consumers. Mathematical models are crucial for exploring the optimal conditions of the drying process for extending the shelf life of food [14]. Accordingly, the changes in the bioactive components and flavor during the drying process are also necessary factors to assess the final quality of dried *Moslae herba*. Nevertheless, there is no comprehensive study investigating the effects of drying kinetics, volatile components and flavor changes, as well as the final quality attributes of *Moslae herba* during the hot air-drying process. Hence, the main objective of this research was to evaluate the effects of different drying temperatures on the drying kinetics, volatile components and changes in flavor of *Moslae herba* during the hot air-drying process. Our study provides an insight into the best conditions of the drying process for obtaining higher-quality *Moslae herba* products.

## 2. Results and Discussion

### 2.1. Drying Kinetics of Moslae herba

The initial moisture content of *Moslae herba* on a dry basis was determined to be 1.5013 ± 0.01 kg water/kg dry matter. The curve of the moisture content on a dry basis (d.b.) against the drying time (h) at different drying temperatures is presented in Figure 1a in order to assess the impact of various drying temperatures (40–60 °C) on the drying kinetics of *Moslae herba*. Our observations revealed that the moisture content of *Moslae herba* decreased gradually as the drying temperature and drying time increased, resulting in a shorter drying time, and the slope of *Moslae herba* moisture content curve became steeper. The *Moslae herba* samples required 32, 15 and 7 h to reach the final moisture content at drying temperatures of 40, 50 and 60 °C, respectively. In other words, the drying time of *Moslae herba* dried at 40 °C and 50 °C was nearly five times and two times that of 60 °C, respectively. This is because the increase in drying temperature increased the difference between the drying air and the partial steam pressure of *Moslae herba* during the process of water removal [15]. These results illustrated that the drying time was determined by the drying temperature.

Moreover, the curve of drying rate against time (h) under different temperatures is shown in Figure 1b. It was obvious that the drying rate initially increased and then decreased during the heating phase. Meanwhile, the drying rate of *Moslae herba* was highest at 60 °C and lowest at 40 °C. This indicates that the higher the drying temperature, the quicker the drying rate. Moreover, the changes in the drying rate were found to agree with our previous studies at different drying temperatures [15,16]. These findings demonstrated that the drying rate could be significantly affected by the drying temperature.

The data presented Table 1 are the statistical results of the seven models. Generally, an optimal fitting condition between the experimental and theoretical data was determined as a higher value of *R*^2^ and a lower value of *χ*^2^ and *RMSE*. These models could be used to describe the drying kinetics of *Moslae herba* at different drying temperatures due to their high coefficient of determination (*R*^2^) values, which ranged from 0.9910 to 0.9976. Among them, Model 1 showed the maximum mean value of *R*^2^ (0.9976) and the minimum mean value of RMSE (0.0120) and *χ*^2^ (2.00 × 10^−4^), indicating that the Midilli model was more accurate for predicting the experimental moisture ratio [17]. In recent years, researchers have also found that the drying characteristics of other food products can be effectively described by the Midilli model [18,19]. Therefore, the Midilli model was selected as the best model for describing the drying behaviors of *Moslae herba* at different drying temperatures. In addition, the parameter values of the Midilli model are presented in Table 2. The values of the drying constant *k* ranged from 0.14540 to 0.65621, and increased with an increase in the drying temperature. This implies that the increase in the drying temperature leads to a steeper drying curve, indicating a faster drying rate [13].

Furthermore, the experimental moisture ratio and predicted values were compared at different hot air-drying temperatures to verify the validity of the selected model. As shown in Figure 1c, it can be observed that these experimental values closely banded around a straight 45° line, indicating that the model was effective in predicting the moisture ratio of *Moslae herba* during the drying process under different temperatures. According to the experimental data, ln*MR* and the corresponding time were regressed and analyzed, and the effective diffusion coefficients were determined to be 3.14 × 10^−5^ m^2^/s, 6.07 × 10^−5^ m^2^/s and 7.39 × 10^−5^ m^2^/s at 40–60 °C, respectively. This might be due to the increased activity of water molecules as the drying temperature increased, resulting in a higher moisture diffusion rate, especially at higher drying temperatures [20]. As shown in Figure 1d, the logarithm of *D_eff_* was a function of the inverse of absolute temperature, and the natural logarithm of the diffusion coefficient (ln*D_eff_*) had a linear relationship with the inverse of temperature (1/T). As can be seen, the correlation coefficient *R^2^* of the regression was 0.9220. Additionally, the effective activation energy (*E_a_*) of water diffusion of *Moslae herba* during the drying process could be estimated to be 37.29 kJ/mol. The larger *E_a_* of the material indicated that it is more difficult to dry, indicating that more starting energy is required in the drying process of *Moslae herba*. Overall, the results above showed that the Midilli model could be used to describe the drying characteristics of *Moslae herba* during the drying process.

### 2.2. Effects of the Drying Process on the Microstructure of Moslae herba

The effects of various drying conditions on microstructure of leaves and stems of *Moslae herba* was observed under scanning electron microscopy (Figure 2). The epidermis layers of *Moslae herba* were covered by hairs or trichomes. After hot air-drying, *Moslae herba* leaves exhibited noticeable changes, such as the severe shrinkage of the underlying epidermal layers and glandular trichomes (Figure 2a,b). Compared with the slight shrinkage of the glandular trichomes of the *Moslae herba* samples dried at 40 °C and 50 °C, the shrinkage of the glandular trichomes of samples dried at 60 °C was stronger. This is because the increasing drying temperature can damage the glandular trichomes [21]. Interestingly, the spirally arranged stems did not show any significant morphological changes during the dehydration process (Figure 2c,d). This is consistent with earlier research [22].

### 2.3. Effects of the Drying Process on the Volatile Compounds of Moslae herba

The flavor attributes of *Moslae herba* were significantly affected by the volatile compounds. The HS-GC-MS technique was used to monitor the alterations in the volatile compounds of *Moslae herba* under different drying temperatures. The diagrams of the total ion flow of all samples with different drying temperatures and the relative content of the volatile compounds of *Moslae herba* samples are shown in Figure 3 and Table 3, respectively. The information on the volatile components was derived from a combination of the NIST20.L database, the retention time and the literature.

The results showed that 13 different volatile compounds and 10 identical compounds were identified in the *Moslae herba* samples. The relative content of the volatile bioactive compounds in the fresh and dried *Moslae herba* samples varied significantly at different drying temperatures, such as thymol, thymol acetate, p-cymene and γ-terpinene. The relative amounts of thymol and thymol acetate derived from the dried samples at the different heating temperatures were significantly increased in comparison with those of fresh samples. Additionally, the relative content of thymol dramatically increased from 28.29% to 56.75%, 55.86% and 55.62% in the dried samples at the different temperatures (Figure 4a). Numerous studies have shown that both thymol and carvacrol are the main flavor attributes of *Moslae herba*. Somewhat differently, carvacrol, as one of major components of *Moslae herba*, was not detected in this study. It has been reported that several factors, such as the environmental conditions, harvest period and extraction method, contribute to the differences in the chemical components of *Moslae herba* [23,24]. Thus, carvacrol was not identified in our study, possibly due to limited content in the original raw material we had obtained. In addition, it has been reported that thymol is the main compound present in thyme, used for centuries as a spice [25,26]. Similarly, the content of thymol acetate increased from 15.91% to 22.64%, 26.46% and 28.38% in the dried samples at different temperatures. A recent study showed that the contents of esters in peppers increased in the early drying process and then decreased with the loss of moisture [27]. It was clear that the volatile esters were greatly affected by the drying temperature during the dehydration process. However, a sharply declining tendency was noticed in the other components during the heating process, including p-cymene (27.77%, 8.40%, 7.46% and 6.39%) and γ-terpinene (10.04%, 1.78%, 1.59% and 1.51%). These results were attributed to the fact that γ-terpinene is easily oxidized to p-cymene by prolonged exposure to air, which continues to be oxidized to thymol [28]. Furthermore, it was reported that some chemical reactions such as enzymatic reactions and oxidation reactions may have occurred at higher temperatures, resulting in a decrease in the variety of compounds and a high content of volatile components [28,29].

Moreover, as shown in Table 3, several volatile compounds in *Moslae herba* such as α-thujene, 3,6,6-trimethyl-bicyclo [3.1.1] hept-2-ene, camphene, α-phellandrene, (R)-isocarvestrene and (−)-β-caryophyllene were only found in fresh *Moslae herba* samples but were not detected in heat-treated samples. Contrary to that, (−)-borneol and (+)-3-carene were only detectable at drying temperatures of 40 °C and 50 °C, respectively. This finding revealed that the amounts of volatile compounds in *Moslae herba* were significantly impacted by the thermal process. The variety in the volatile components of the fresh samples was more than that in dried samples, but the amounts of the main volatile components were less than in the dried samples. Overall, the results proved that raising the temperature of drying could promote the release of some volatile bioactive components in *Moslae herba*, indicating that higher heating temperatures may have a potential effect on the flavor properties of dried *Moslae herba*. The PCA of the volatile components of the fresh *Moslae herba* and the *Moslae herba* dried under different drying temperatures is presented in Figure 4b. It was obvious that the dried *Moslae herba* under different temperatures had similar volatile components and was clearly distinguished from the fresh group. This is consistent with the results we obtained previously.

### 2.4. Effect of the Drying Process on the Aroma Profile of Dried Moslae herba

The electronic nose technology (E-nose) has been widely known to be sensitive to the aroma of various products. Since it is able to detect slight changes in volatile compounds, which might be responsible for the different responses of the sensor. Moreover, it is an excellent method for analyzing aromas and has been widely used in the fields of biomedicine, environment, manufacturing, food, pharmaceuticals and other scientific research fields [30]. To assess the influence of various drying treatments on the aroma profile of *Moslae herba*, an E-nose equipped with 14 sensors was adopted in this study to analyze the combined aroma characteristics of *Moslae herba*. The sensitivity of the volatile component sensors to *Moslae herba* was approximately the same for the different drying temperatures, but the intensity of the response values varied. This is because the intensity of the response is determined by the variation in the content of volatile components. The radar fingerprint of the aroma distribution in *Moslae herba* is presented in Figure 5a, in which all *Moslae herba* samples presented relatively higher values in S1 (sensitive to aromatic compounds) and S8 (sensitive to amines), followed by S2 (sensitive to nitrogen oxides), S6 (sensitive to thionine) and S12 (sensitive to sulfides from environment). Whereas, sensors S3 (sensitive to sulfides from vegetables), S7 (sensitive to combustible gas) and S14 (sensitive to volatile gases from cooking) showed smaller response values. Meanwhile, it can be clearly observed that notable differences in the signal intensity of the sensors between the fresh samples and dried samples, indicating notable alterations in the composition of the volatile compounds during the heating process. Among them, the signal intensity of S1 (sensitive to aromatic compounds) and S8 (sensitive to amines) was significantly higher in fresh samples as compared with dried samples, and they decreased with a rising temperature, which may be due to the loss of aromatic compounds at higher temperatures. In a previous study, researchers used electronic nose technology to monitor the changes in odor of *Flammulina velutipes* during the hot air-drying process, which was similar to our results [31]. Therefore, it is feasible to use electronic nose technology to distinguish the variations in the odor of *Moslae herba* during the hot air-drying process.

PCA was used to analyze the sensory evaluation data in our study to underline the variations in the aroma profile of *Moslae herba* at various drying temperatures. Generally, the first two PCs (PC1 and PC2) accounted for 85%, suggesting the feasibility of the method [32]. Figure 5b displays the PCA of the aromatic compounds in the *Moslae herba* samples. PC1 and PC2 accounted for 72.7% and 13.3%, respectively, with a cumulative contribution of 86.0% (more than 85%), which indicated that these two feature components mainly contributed to the sensory attributes of *Moslae herba* under different drying conditions. The fresh *Moslae herba* samples and the samples dried at different temperatures were in a relatively independent area and clearly separate from each other, indicating that the volatile components presented in these groups are clearly different. The differences may have resulted from the production of new volatile substances and similar volatile substances during the hot air-drying process. *Moslae herba* samples dried at 50 °C showed a clear overlap with samples dried at 60 °C, indicating similar flavor compounds being present in these two samples, which was in agreement with the results of the HS-GC-MS analysis. These findings clearly suggested that the PCA of the E-nose data could be used to distinguish the flavor and sensory attributes of *Moslae herba* under different drying conditions.

### 2.5. Total Phenolic and Flavonoid Contents

It has been reported that natural phenolic compounds are extensively present in edible plants and are thought to exert their beneficial health functions primarily due to their antioxidant capacity [33,34]. Flavonoids are the predominant phenolic compounds in *Moslae herba*. As shown in Figure 6a, TPC varied notably among the tested *Moslae herba* samples, ranging from 9.88 to 13.99 mg GAE/g dry weight. Fresh samples contained the highest TPC at 13.99 mg GAE/g dry weight, followed by samples dried at 60 °C at 13.36 mg GAE/g dry weight. Moreover, fresh *Moslae herba* had the highest value of TFC at 54.34 mg RE/g dry weight, followed by the samples dried at 60 °C (47.58 mg RE/g dry weight) (Figure 6b). These results demonstrated that different heating treatments had a significant impact on the content of bioactive components derived from *Moslae herba*, and appropriate high drying temperatures are beneficial for the retention of bioactive components due to the short drying time.

### 2.6. Antioxidant Activities

In general, polyphenolic compounds are considered to be antioxidants with good antioxidant potential. Higher antioxidant activity may be attributed to higher levels of phenols and flavonoids [35]. The results of the DPPH assay are depicted in Figure 6c, with the values ranging from 5.44 to 10.19 mmol TE/g dry weight, and the highest antioxidant activity being almost twice as high as the lowest. The FRAP values measured for the reducing power was in the range of 2.28 and 3.77 mmol Fe^2+^/g dry weight (Figure 6d). It can be observed that fresh *Moslae herba* samples exhibited higher DPPH and FRAP antioxidant activity compared with the dried samples. In addition, the antioxidant capacity of *Moslae herba* samples after drying at 60 °C was the strongest among all of the dried samples at various temperatures, but after the fresh group. This could be explained by higher phenolic and flavonoid contents being present in fresh *Moslae herba* and in *Moslae herba* dried at 60 °C.

## 3. Materials and Methods

### 3.1. Raw Material and Chemical Regents

Fresh *Moslae herba* samples were obtained from Jiangxi Fenyi Jinbei Medicinal Materials Co., Ltd. (Xinyu, China) and stored at 4 °C before the experiment.

1,1-Diphenyl-2-picrylhydrazyl (DPPH) was procured from Tixiai Huacheng Industrial Development Co., Ltd. (Shanghai, China), 2,4,6-tripyridyl-s-triazine (TPTZ) and Trolox were purchased from Shanghai Yuanye Biochemical Technology Co., Ltd. (Shanghai, China). The rest of the reagents were analytically pure.

### 3.2. Hot Air-Drying of Moslae herba

A batch of fresh *Moslae herba* (400 g) cut into sections was spread out evenly (2~3 cm thick) on a tray and dried in a hot air blast drying oven at temperatures of 40 °C, 50 °C and 60 °C (BPG-9070A, Shanghai Yiheng Scientific Instrument Co., Ltd., Shanghai, China). As shown in Figure 7, heat was produced by the heater on the wall of the oven, and hot air circulation was used to dry the *Moslae herba* samples. The hot air entered the air inlet and was sent out from the air outlet at a speed of 2.0 m/s. The tray was weighed on an electronic scale (YH-A20002A, Ruian Yingheng Electric Co., Ltd., Ruian, China) at intervals of 5 min (0–30 min), 10 min (30–60 min), 20 min (60–120 min), 30 min (120–180 min) and 60 min until a constant weight was reached.

### 3.3. Determination of the Moisture Content

Fresh *Moslae herba* was placed in a moisture meter at 105 °C and baked to a constant weight (DSH-50-1, Yue Ping Scientific Instruments Co., Ltd., Shanghai, China), and the weight loss was computed as a percentage of the initial weight. Moisture content (*M*) was calculated using Equation (1).
(1)$$M=\frac{W_{w}-W_{d}}{W_{d}}$$
where *M* is the moisture content of the *Moslae herba*, *W_w_* is the wet weight of *Moslae herba* (g) and *W_d_* is the dry weight of *Moslae herba* (g). The experiments were carried out three times.

### 3.4. Determining the Behavior of the Hot Air-Drying Process

#### 3.4.1. Determination of the Moisture Ratio

The moisture ratio (*MR*) is the residual moisture of the material under specific drying conditions, which can be a good indicator of the drying speed at different drying temperatures. The moisture ratio of *Moslae herba* in the hot air-drying experiments was calculated according to Equation (2)
(2)$$MR=\frac{M_{t}{\;-\;M}_{e}}{M_{0}-{\;M}_{e}}$$
where *M*_0_, *M_t_* and *M_e_* are the initial moisture content, the moisture content at time *t* and the equilibrium moisture content of *Moslae herba*, respectively. Equation (2) can be simplified as follows [22,36]:(3)$$MR=\frac{M_{t}}{M_{0}}$$

#### 3.4.2. Determination of the Drying Rate

The drying rate (*DR*) is the mass of moisture vaporized per unit area (contact area between material and drying medium) of wet material per unit time, as calculated using Equation (4) [37].
(4)$$DR=\frac{-{(M}_{t+\Delta t}{\;-\;M}_{t})}{\Delta t}$$
where *M_t+_*_∆*t*_ is the moisture content of *Moslae herba* at time *t* + ∆*t* (min).

#### 3.4.3. Analysis of Drying Behaviors of *Moslae herba*

To investigate the drying characteristics of *Moslae herba* and describe the drying process of *Moslae herba* as accurately as possible, seven empirical models (listed in Table 4) were used to select the most appropriate model for describing the drying characters of *Moslae herba*. The goodness of fit was evaluated on the basis of the coefficient of determination (*R*^2^), the root mean square error (*RMSE*) and the chi-square (*χ*^2^). The higher the *R*^2^ values, and the lower the *χ*^2^ and *RMSE* values, the better the goodness of fit [38,39]. These parameters were calculated as follows
(5)$$R^{2}=1\;-\;\frac{\sum_{i=1}^{N}{{(\mathrm{MR}}_{\exp,i}-{\;\mathrm{MR}}_{\mathrm{pre},i})}^{2}}{\sum_{i=1}^{N}{{(\mathrm{MR}}_{\exp,i}-\bar{{\mathrm{MR}}_{\mathrm{pre},i}})}^{2}}$$
(6)$$\chi^{2}=\frac{\sum_{i=1}^{N}{{(\mathrm{MR}}_{\exp,i}-{\mathrm{MR}}_{\mathrm{pre},i})}^{2}}{N\;-\;Z}$$
(7)$$\mathrm{RMSE}=\sqrt{\frac{1}{N}\sum_{i=1}^{N}{{(\mathrm{MR}}_{\exp,i}-{\;\mathrm{MR}}_{\mathrm{pre},i})}^{2}}$$
where *MR_exp_*,*_i_* is the *i*th experimental moisture ratio, *MR_pre_*,*_i_* is the *i*th predicted moisture ratio, *N* is the number of observations and *Z* is the number of constants.

#### 3.4.4. Effective Diffusion Coefficient (Deff)

The effective diffusivity (*D_eff_)* was computed according to Fick’s diffusion model. During the hot air-drying process of *Moslae herba*, the internal moisture flow of *Moslae herba* is usually considered to be by diffusion (liquid or steam). The *D_eff_* value of moisture during the drying of *Moslae herba* can be calculated using Equation (8)
(8)$${\mathrm{MR}=8\pi }^{-2}\sum_{i=1}^{N}{\left(2n+1\right)}^{-2}\exp[-{\left(2n+1\right)}^{2}\pi^{2}D_{\mathrm{eff}}{\mathrm{tL}}_{0}{}^{-2}]$$
where *t* is the experimental time (s), *L* is the thickness of the thin-layer *Moslae herba* experimental sample (m) and *n* is the number of experimental samples.

For long drying periods, the limiting form of the equation above is obtained by considering only the first (*n* = 1) term of the series and is expressed in logarithmic form. Equation (8) can be simplified as follows:(9)$${\mathrm{lnMR}=\ln(8/\pi }^{2})\;-\frac{\pi^{2}D_{\mathrm{eff}}t}{L_{0}{}^{2}}$$

#### 3.4.5. Activation Energy (*E_a_*)

The activation energy (*E_a_*) was calculated using a modification of the Arrhenius equation. The activation energy (*E_a_*) represents the starting energy required to evaporate a unit of moles of water in the drying process. Generally, a larger *E_a_* for the material indicates that it is more difficult to dry. The relationship between *D_eff_* and *E_a_* can be established according to the Arrhenius equation [46,47]
(10)$$D_{\mathrm{eff}}{=D}_{0}\exp(-\frac{E_{a}}{\mathrm{RT}})$$
where *D_eff_* is the coefficient of effective moisture diffusion (m^2^/s), *D*_0_ is the pre-exponential factor of the Arrhenius equation (m^2^/s), *E_a_* is the activation energy (kJ/mol), *R* is the universal gas constant (8.314 kJ/mol) and *T* is the absolute temperature (K). The activation energy (*E_a_*) can be determined by linearizing with ln(*D_eff_*) versus 1/T (Equation (10)).

### 3.5. Scanning Electron Microscopy

Changes in the microstructure of the stems and leaves of *Moslae herba* due to the drying process were examined using a scanning electron microscope (SEM; FEI-Quanta250, Waltham, MA, USA). After gold plating of the surfaces of the samples in a vacuum, SEM images of *Moslae herba* samples were obtained at an acceleration voltage of 20 kV.

### 3.6. Headspace Gas Chromatography–Mass Spectrometry (HS-GC–MS) Analysis

The volatile components of *Moslae herba* were analyzed using a HS-GC–MS apparatus (8860-5977, Agilent, Palo Alto, CA, USA). Fresh and dried *Moslae herba* samples (0.5 g) were placed in a 20 mL glass vial. The volatile compounds were separated by using an HP-5MS type column (30 m × 0.25 mm × 0.25 μm). Helium was used as the carrier gas with a flow rate of 1 mL/min and a split ratio of 50:1. The inlet temperature was 250 °C and the injection volume was 0.2 μL. The heating procedure was as follows. The oven program began at 40 °C held for 2 min, increased to 60 °C at 5 °C/min and held for 3 min, raised to 150 °C at 5 °C/min and held for 3 min, and finally increased to 250 °C at 25 °C/min for 3 min. The temperatures of the ion source and transfer line were 230 °C and 280 °C, respectively. The mass spectra were acquired by the electron impact at an energy of 70 eV, and the data were collected at 1/scan over the mass range *m*/*z* 35–550. The identified components were analyzed by a search of the mass spectrometry database (NIST20.L). The quantification of volatile compounds was conducted using the normalized peak area measurements. Each *Moslae herba* sample was measured three times, and the mean value was taken for further analysis.

### 3.7. Determination of the Flavor Characteristics of Moslae herba

The flavor characteristics of *Moslae herba* were determined using an I-Nose type electronic nose (Ruifen Intelligent Technology Co., Ltd., Shanghai, China). The sensors used for this analysis are shown in Table 5. For this, 1 g of *Moslae herba* powder at different drying temperatures was placed into a 20 mL glass vial and incubated for 30 min at room temperature. In line with the headspace aspiration method, the sample of powder was directly placed in a vial and then a syringe needle was inserted into the headspace above the sample to collect the flavor compounds. Each group of samples was equilibrated 3 times. The injection needle was inserted into the vial, each test time was set to 120 s, and the sensor flushing time was set to 180 s.

### 3.8. Determination of the Bioactive Ingredients in Moslae herba

#### 3.8.1. Preparation of the Extracts 

About 0.1 g of the *Moslae herba* samples was mixed with 2.5 mL of 60% ethanol, and then an ultrasonic examination was conducted in an ultrasound bath for 40 min at 70 °C (KQ5200DA, Kunshan Ultrasonic Instrument Co., Ltd., Shanghai, China). The extracts were centrifuged for 10 min at 4000 rpm (TGL-16B, Shanghai Anting Scientific Instrument Factory, Shanghai, China), and then the supernatants were collected. The supernatants were diluted 100 times to determine the antioxidant capacity.

#### 3.8.2. Determination of the Total Phenolic Content (TPC)

The TPC of *Moslae herba* was measured by the Folin–Ciocalteu assay according to [48] with some modifications. For this, 25 μL of a gallic acid standard (31.25–500 μg/mL) or the *Moslae herba* extracts was mixed with 125 μL of Folin’s phenol solution (0.2 M). After 10 min, a 125 μL saturated Na_2_CO_3_ solution (10 g/100 mL) was added to the mixture. The absorbance was determined at 765 nm using UV spectrophotometry (Molecular Devices, San Jose, CA, USA) after 30 min of reaction in the dark. The results were expressed as gallic acid equivalents (GAE, mg/g of dry weight).

#### 3.8.3. Determination of the Total Flavonoid Content (TFC)

The TFC of *Moslae herba* was determined using a colorimetric method [33] with slight modifications. For this, 25 μL of a rutin standard (62.5–1000 μg/mL) or the *Moslae herba* extracts was mixed with 110 μL of NaNO_2_ (0.066 M). After 30 min, 15 μL of AlCl_3_ (0.75 M) was added to the mixture and allowed to stand for 6 min before 100 μL of a NaOH solution (0.5 M) was added. The absorbance was read at 510 nm using UV spectrophotometry (Molecular Devices, CA, USA). The results were expressed as rutin equivalents (RE, mg/g of dry weight).

### 3.9. Antioxidant Ability

#### 3.9.1. DPPH Radical Scavenging Assay

The DPPH assay of *Moslae herba* was measured using the method of [49]. For this, 25 μL of the extracts or a Trolox standard (62.5–1000 μM) was mixed with 200 μL of a methanolic solution of DPPH (350 μM). After 6 h of reaction in the dark at room temperature, the absorbance value was determined at 517 nm. The results were expressed as Trolox equivalents (TE, mg/g of dry weight).

#### 3.9.2. Ferric Reducing Antioxidant Power (FRAP) Assay

The FRAP assay of *Moslae herba* was performed using the method of [50] with minor modifications. Briefly, we prepared a FRAP working solution by mixing a sodium acetate buffer solution (0.3 M), a ferric chloride solution (20 mM) and a TPTZ (10 mM) solution at a ratio of 10:1:1. Then 10 μL of the *Moslae herba* extracts were mixed with 180 μL of the FRAP working solution preheated to 37 °C for 10 min, and the absorbance value was determined at 593 nm. Six concentrations (4.12–1000 μg/mL) were used to prepare the standard curve of FeSO_4_·7H_2_O. The results were expressed as mmol FeSO_4_ (Fe^2+^)/g dry weight.

### 3.10. Statistical Analysis

The experimental results were analyzed using Origin 2019b software (Origin Lab, Northampton, MA, USA). The PCA was carried out by the Bioinformatics platform (https://www.bioinformatics.com.cn/, accessed on 26 April 2023). The one-way analysis of variance (ANOVA) was performed by Tukey’s test (*p* < 0.05). All samples had three replicates, and the data were reported as the mean ± SD.

## 4. Conclusions

In the current study, *Moslae herba* was dried using thin-layer hot air-drying at different temperatures (40 °C, 50 °C and 60 °C). The findings revealed that a shorter drying time was required for *Moslae herba* at a higher drying temperature. Meanwhile, the Midilli model was found to have the best fit for describing the drying process of *Moslae herba* among the seven selected models. In addition, the changes in the microstructure and flavor characteristics of *Moslae herba* under different drying conditions were analyzed using SEM, HS-GC-MS and E-nose techniques. This revealed that the shrinkage of glandular trichomes was detected more significantly with increasing processing temperatures during the drying process. Moreover, 23 volatile compounds in total were identified in the *Moslae herba* samples, and the majority of them decreased after hot air-drying but some of these volatile substances increased. Thymol, the main characteristic component of *Moslae herba*, increased from 28.29% in fresh samples to 56.75% at 40 °C, 55.86% at 50 °C and 55.62% at 60 °C. In contrast, other volatile compounds such as p-cymene and γ-terpinene reduced after drying. This indicates that a moderate increase in the drying temperature during hot air-drying may contribute to enhancing the aroma profile of dried *Moslae herba*. According to the E-nose analysis, fresh *Moslae herba* samples were clearly distinct from the dried *Moslae herba* samples, and less negative effects on the volatile composition and associated flavor qualities of the *Moslae herba* samples with lower temperatures. Among the dried samples, drying at 60 °C led to favorable effects on the total phenolic and total flavonoid contents of *Moslae herba* and its antioxidant capacity as compared with the other drying temperatures, indicating that appropriate high temperatures are beneficial for the retention of bioactive compounds. The results from this work provide a theoretical foundation for the development of drying technologies and related products for *Moslae herba*. To sum up, a new idea based on the concept that “quality comes from technology” has been established, which provided a comprehensive description of the factors affecting the quality of *Moslae herba* during the drying process and was used to evaluate the optimal drying process. Future research should design a standardized and digital quality control system for aromatics in traditional Chinese medicine, build an expert system and develop intelligent and automatic processing equipment.in order to improve the drying efficiency, physicochemical properties and sensory quality of the product.

## Figures and Tables

**Figure 1 molecules-28-03898-f001:**
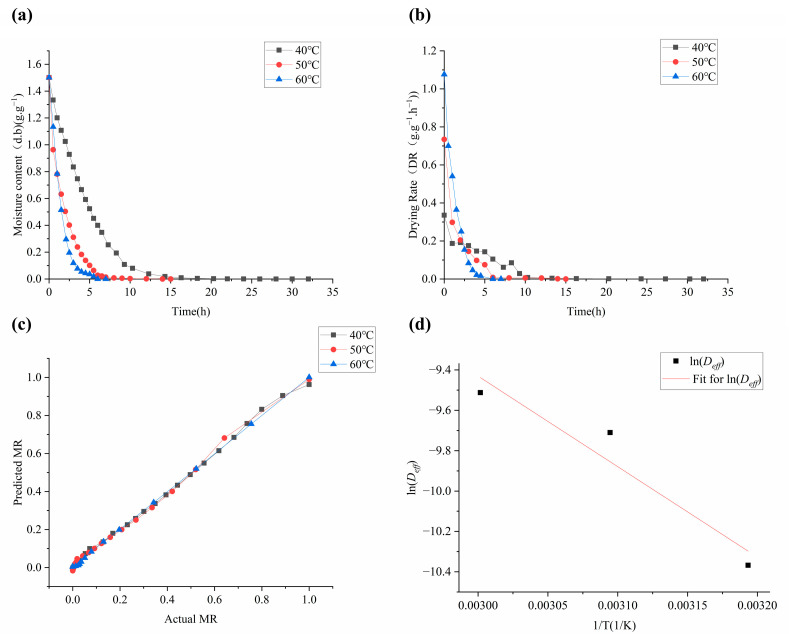
Drying kinetic curves of *Moslae herba*. (**a**) Moisture content (dry basis) curves of *Moslae herba* at temperatures of 40 °C, 50 °C and 60 °C. (**b**) Drying rate curves of *Moslae herba* at temperatures of 40, 50 and 60 °C. (**c**) Relationship between the predicted and measured values of the moisture ratio. (**d**) Curve of the relationship between ln*Deff* and 1/T.

**Figure 2 molecules-28-03898-f002:**
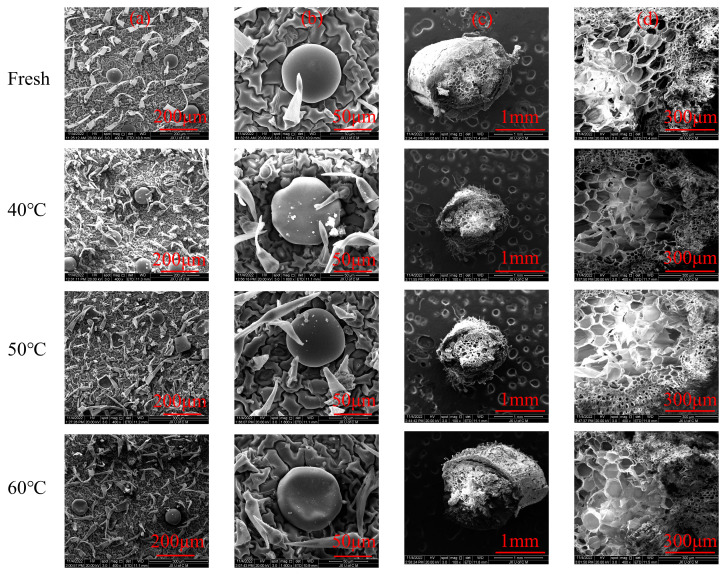
The microstructural images of fresh and dried *Moslae herba* samples. (**a**) Epidermal surface of the leaves under 400× magnification. (**b**) Epidermal surface of the leaves under 1600× magnification. (**c**) Cross-sectional view of the stems under 100× magnification. (**d**) Cross-sectional view of the stems under 400× magnification. Fresh, fresh *Moslae herba*; 40 °C, *Moslae herba* after hot air-drying at 40 °C; 50 °C, *Moslae herba* after hot air-drying at 50 °C; 60 °C, *Moslae herba* after hot air-drying at 60 °C.

**Figure 3 molecules-28-03898-f003:**
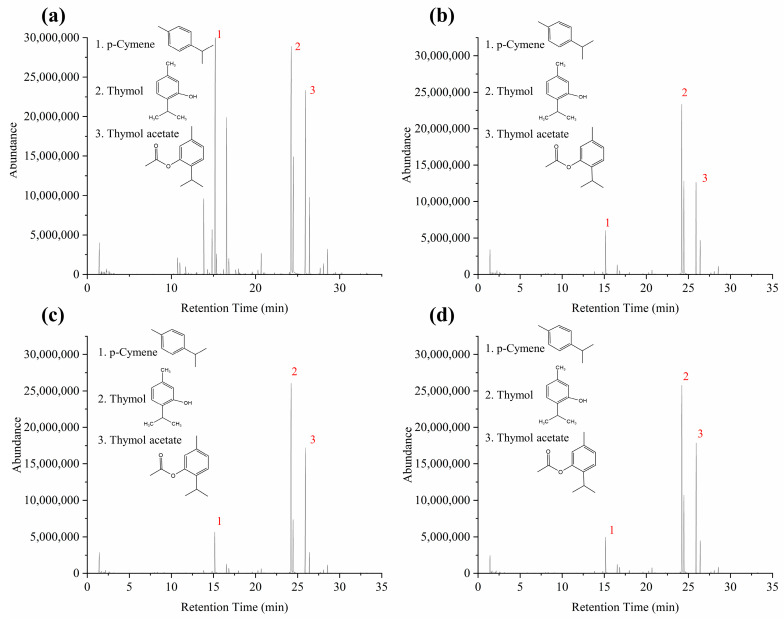
HS-GC-MS total ion flow diagram of the volatile components of *Moslae herba* at different drying temperatures. (**a**) Fresh *Moslae herba*, (**b**) *Moslae herba* after drying at 40 °C, (**c**) *Moslae herba* after drying at 50 °C and (**d**) *Moslae herba* after drying at 60 °C.

**Figure 4 molecules-28-03898-f004:**
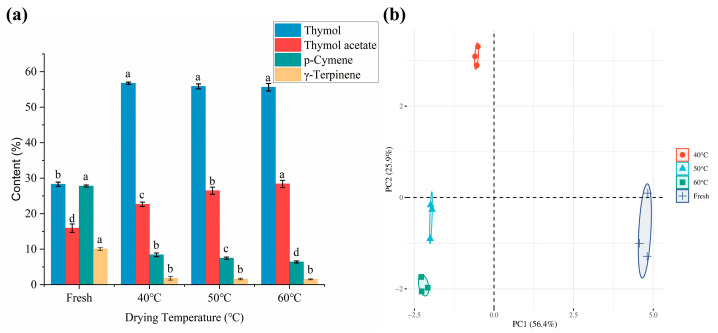
(**a**) The contents of four common components in *Moslae herba* at different temperatures. (**b**) PCA of the HS-GC-MS of *Moslae herba* volatile components under different drying temperatures. Bars with different letters represent significant difference (*p* < 0.05).

**Figure 5 molecules-28-03898-f005:**
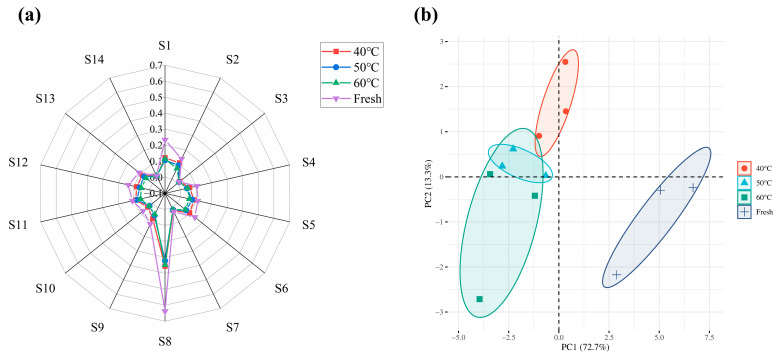
Radar chart (**a**) and principal component analysis (**b**) of the E-nose analysis of fresh *Moslae herba* and dried *Moslae herba* samples under different drying temperature conditions.

**Figure 6 molecules-28-03898-f006:**
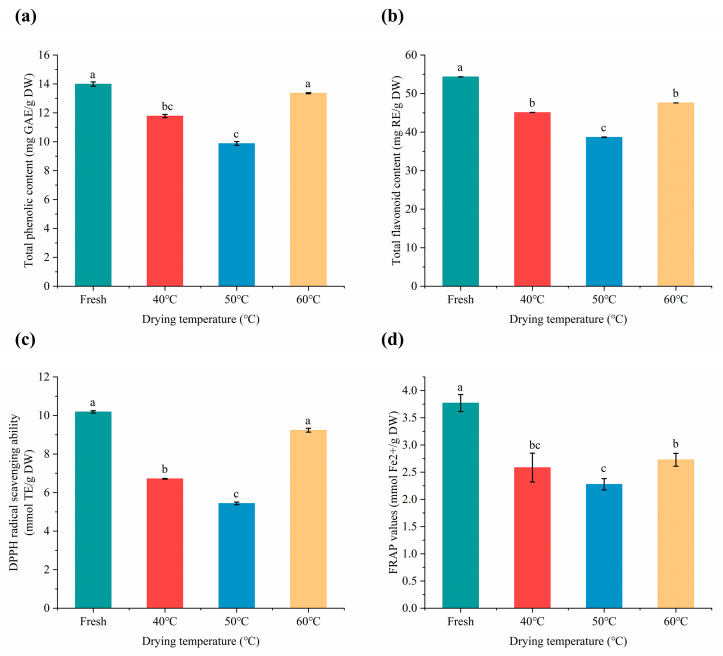
Effects of different drying temperatures on content of TPC (**a**), TFC (**b**), DPPH (**c**) and FRAP (**d**) in *Moslae herba*. Bars with different letters represent significant differences (*p* < 0.05).

**Figure 7 molecules-28-03898-f007:**
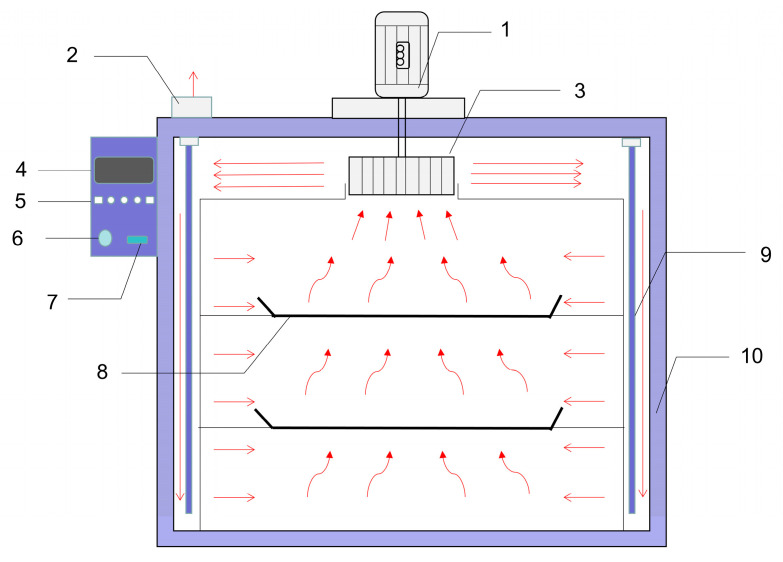
Schematic diagram of the hot air-drying system: 1, blower motor; 2, air outlet; 3, air inlet; 4, display; 5, operation buttons; 6, wind speed adjustment button; 7, switch; 8, tray; 9, heater; 10, insulation layer.

**Table 1 molecules-28-03898-t001:** Fitting results of seven models.

Model	SP	40 °C	50 °C	60 °C	MSP
1	*R* ^2^	0.9977	0.9954	0.9996	0.9976
	*χ* ^2^	0.0002	0.0003	0.0000	0.0002
	RMSE	0.0142	0.0164	0.0053	0.0120
2	*R* ^2^	0.9969	0.9934	0.9996	0.9966
	*χ* ^2^	0.0003	0.0005	0.0000	0.0003
	RMSE	0.0142	0.0209	0.0056	0.0136
3	*R* ^2^	0.9977	0.9933	0.9996	0.9969
	*χ* ^2^	0.0056	0.0080	0.0005	0.0047
	RMSE	0.0144	0.0205	0.0056	0.0135
4	*R* ^2^	0.9969	0.9934	0.9996	0.9966
	*χ* ^2^	0.0003	0.0005	0.0000	0.0003
	RMSE	0.0142	0.0209	0.0056	0.0136
5	*R* ^2^	0.9972	0.9957	0.9997	0.9975
	*χ* ^2^	0.0003	0.0003	0.0000	0.0002
	RMSE	0.0160	0.0169	0.0048	0.0126
6	*R* ^2^	0.9926	0.9869	0.9934	0.9910
	*χ* ^2^	0.0007	0.0010	0.0006	0.0008
	RMSE	0.0267	0.0303	0.0239	0.0270
7	*R* ^2^	0.9947	0.9894	0.9949	0.9930
	*χ* ^2^	0.0128	0.0126	0.0062	0.0105
	RMSE	0.0218	0.0258	0.0196	0.0224

Note: SP refers to the statistical parameters, and MSP represents the average value of the statistical parameters at different temperatures.

**Table 2 molecules-28-03898-t002:** Coefficients of Model Number 1 at different drying temperatures.

Temperature	40 °C	50 °C	60 °C
*a*	0.96247	0.98966	1.00079
*k*	0.14540	0.64514	0.65621
*y*	1.22734	0.79928	1.21805
*b*	−2.4287 × 10^−4^	−0.00339	4.19041 × 10^−4^

Note: *a*, *k*, *b* and *y* are the drying constants.

**Table 3 molecules-28-03898-t003:** Volatile components of *Moslae herba* at different drying temperatures as determined by HS-GC-MS.

No	t_R_ (min)	Formula	Compounds	Common Name	Fresh	40 °C	50 °C	60 °C
1.	1.424	/	Unknown	/	1.54	4.36	3.69	3.45
2.	2.286	C_2_H_4_O_2_	Acetic acid	/	0.34	0.62	/	/
3.	10.733	C_10_H_16_	5-Isopropyl-2-methylbicyclo[3.1.0]hex-2-ene	α-Thujene	1.09	/	/	/
4.	11.010	C_10_H_16_	3,6,6-Trimethyl-bicyclo [3.1.1] hept-2-ene	/	0.82	/	/	/
5.	11.696	C_10_H_16_	Camphene	/	0.53	/	/	/
6.	13.841	C_10_H_16_	β-Myrcene	7-Methyl-3-methyleneocta-1,6-diene	4.59	0.58	0.53	/
7.	14.302	C_10_H_16_	α-Phellandrene	/	0.32	/	/	/
8.	14.832	C_10_H_16_	1-methyl-4-(1-methylethyl)-1,3-Cyclohexadiene	α-Terpinene	3.14	0.49	/	/
9.	15.157	C_10_H_14_	p-Cymene	/	27.77	8.40	7.46	6.39
10.	15.344	C_10_H_16_	(R)-1-methyl-5-(1- methylethenyl)-cyclohexe (R)-isocarvestrene	1.07	/	/	/	
11.	16.199	C_10_H_16_	β-Ocimene	/	0.32	/	/	/
12.	16.568	C_10_H_16_	γ-Terpinene	/	10.04	1.78	1.59	1.51
13.	17.114	C_10_H_18_O	(1α,2α,5α)-2-methyl-5-(1-methylethyl)bicyclo[3.1.0]hexan-2-ol	(E)-sabinene hydrate	1.15	1.16	0.91	1.06
14.	17.970	C_10_H_16_	(+)-3-Carene	/	/	/	/	0.50
15.	20.281	C_10_H_18_O	Borneol	/	0.31	0.42	0.58	0.50
16.	20.667	C_10_H_18_O	Terpinen-4-ol	/	1.07	0.77	0.78	0.88
17.	24.211	C_10_H_14_O	Thymol	/	28.29	56.75	55.86	55.62
18.	24.783	C_10_H_18_O	(1Sendo)-1,7,7-trimethyl-bicyclo[2.2.1]heptan2-ol	(−)-Borneol	/	0.44	/	/
19.	26.170	C_12_H_16_O_2_	Phenol 5-methyl-2-(1-methylethyl) acetate	Thymol acetate	15.91	22.64	26.46	28.38
20.	27.679	C_15_H_24_	Caryophyllene	(−)-β-Caryophyllene	0.33	/	/	/
21.	28.065	C_15_H_24_	2,6-Dimethyl-6-(4-methyl-3-pentenyl)-bicyclo[3.1.1]hept-2-ene	α-Bergamotene	0.52	0.50	0.54	0.45
22.	28.552	C_15_H_24_	Humulene	(1E,4E,8E)-α-humulene	1.35	1.51	1.45	1.07
23.	33.222	/	Unknown	/	/	/	/	0.47

Note: / means not detected.

**Table 4 molecules-28-03898-t004:** Models equations for the thin-layer drying curve.

No	Model Name	Model Equation	References
1	Midilli	*MR* = *a* exp(*−kt^y^*) + *bt*	[40]
2	Page	*MR* = exp(*−kt^y^*)	[41]
3	Modified Page	*MR* = *a* exp[*−*(*kt^y^*)]	[42]
4	Overhults	*MR =* exp[*−*(*kt*)*^y^*]	[43]
5	Two-term exponential	*MR* = *a* exp(*−kt*) + (1 − *a*) exp(*−kat*)	[44]
6	Newton	*MR* = exp(*−kt*)	[45]
7	Logarithmic	*MR* = *a* exp(*−kt*) + *c*	[46]

Note: *t*, drying time (s); *a*, *k*, *b*, *y*, *c*, model coefficients.

**Table 5 molecules-28-03898-t005:** Sensors and the types of flavor detected by the E-nose.

Sensors	Types of Flavor Detected
S1	Aromatic compounds
S2	Nitrogen oxides
S3	Sulfides from vegetables
S4	Organic acids and terpenoids
S5	Biosynthetic compounds
S6	Thionine
S7	Combustible gas
S8	Amines
S9	Hydrogen
S10	Hydrocarbons
S11	Volatile organic compounds
S12	Sulfides from the environment
S13	Ethylene
S14	Volatile gases from cooking

## Data Availability

Data are contained within the article.

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
