# Peer review of "Effect of Drying Kinetics, Volatile Components, Flavor Changes and Final Quality Attributes of Moslae herba during the Hot Air Thin-Layer Drying Process"

_molecules, 2023, doi:10.3390/molecules28093898_

Round 1
Reviewer 1 Report
The manuscript presents interesting findings on the effect of different temperatures on the drying characteristics, textural properties, bioactive compounds, flavor changes and final quality attributes of Moslae Herba during the hot air-drying process. The overall results elucidated that drying Moslae Herba at the temperature of 60°C efficiently enhanced the final quality by significantly reducing the drying time and maintaining the bioactive compounds. The study is well-designed and the results are presented clearly. However, addressing the issues mentioned below will enhance the scientific rigor and clarity of the manuscript and make it more impactful.
1. The language of the manuscript should be revised with some native English-speaking person.
2. The authors should provide a detailed description of the methods used to allow the readers to better understand the study's methodology and interpret the results.
3. The first appearance of abbreviations needs to be marked with complete definitions.
4. Figures should be self-explanatory. Please improve figure legends
5. Readers may appreciate some discussion of the limitations of the current study and used methodology.
6. Enrich the pool of references with recent citations. Support the first statement of section 3.5 “It is reported that natural phenolic compounds are extensively presented in edible plants and are thought to exert their beneficial health functions primarily due to their antioxidant capacity” with the following references: doi: 10.1016/j.plaphy.2022.06.021
7. The study’s authors could consider including a section on future research directions/future perspectives at the end of the conclusion.
Author Response
Response to Reviewer 1 Comments
Point 1: The language of the manuscript should be revised with some native English -speaking person.
Response 1: We appreciated for your valuable comment. In order to accurately introduce the research content of this manuscript, we have invited some native English-speaking person to carefully revise some language and writing problems. We really hope that the language level will be substantially improved.
Point 2: The authors should provide a detailed description of the methods used to allow the readers to better understand the study's methodology and interpret the results.
Response 2: Thank you for the valuable comment. We have added the method in detail in Section 2.2 (page 3, line 99-100) and Section 2.7 (page 6, line 191-193).
Point 3: The first appearance of abbreviations needs to be marked with complete definitions.
Response 3: We thank you for pointing out this issue. I have supplemented the complete definitions of SEM (page 1, line 25), HS-GC-MS (page 5, line 172) and PCA (page 1, line 25) in the article, respectively.
Point 4: Figures should be self-explanatory. Please improve figure legends.
Response 4: Thank you for pointing out this issue. We have described the legend in detail, such as Figure 4 and Figure 7.
Point 5: Readers may appreciate some discussion of the limitations of the current study and used methodology.
Response 5: Thanks to your important comment. We have already described the limitations of the current study and used methodology in the conclusion (page 17, line 478-481).
Point 6: Enrich the pool of references with recent citations. Support the first statement of section 3.5 “It is reported that natural phenolic compounds are extensively presented in edible plants and are thought to exert their beneficial health functions primarily due to their antioxidant capacity” with the following references: doi: 10.1016/j.plaphy.2022.06.021.
Response 6: Thanks to your valuable suggestion. We read the article recommended by the reviewer and found a lot to learn from it and cited it in the article (page 15, line 431).
Point 7: The study’s authors could consider including a section on future research directions/future perspectives at the end of the conclusion.
Response 7: Thanks to the reviewer's suggestion. We have added to the conclusion an insight into optimizing the processing of Moslae Herba (page 17, line 481-483).

Reviewer 2 Report
The present research was carried to investigate the effect of different temperatures (40, 50 and 60℃) on the drying characteristics, textural properties, bioactive compounds, flavor changes and final quality attributes of Moslae Herba during the hot air-drying process
The topic is relevant in the field. To date, no comprehensive study investigating the effects of drying kinetics,volatile components and flavor changes as well as the final quality attributes of Moslae Herba during the hot air-drying process.
No specific imporvement is needed.
There are the conclusions consistent with the evidence.
There are the references appropriate.
The figures and tables are appropriate.
Author Response
Response to Reviewer 2 Comments
Point: The present research was carried to investigate the effect of different temperatures (40, 50 and 60℃) on the drying characteristics, textural properties, bioactive compounds, flavor changes and final quality attributes of Moslae Herba during the hot air-drying process
The topic is relevant in the field. To date, no comprehensive study investigating the effects of drying kinetics, volatile components and flavor changes as well as the final quality attributes of Moslae Herba during the hot air-drying process.
No specific imporvement is needed.
There are the conclusions consistent with the evidence.
There are the references appropriate.
The figures and tables are appropriate.
Response: Thank you very much for tanking your time to review this manuscript. We are grateful to have your approval.

Reviewer 3 Report
This article compares the effects of different temperatures on the drying characteristics, texture properties, bioactive components, flavor changes, and final quality attributes of Moslae Herba during hot air drying. There are several questions about this article.
1. Why choose 40 ℃, 50 ℃, and 60 ℃ for subsequent research?
2. Please unify the font size. For example, the font on formula three.
3. There are formatting errors in the text, such as ‘ofMoslae Herba’ in the title without spaces, and the formula needs to be aligned.
4. Please note the unit corresponding to each letter in the formula.
5. What are the drying constants a, k, y, and b in Table 4?
6. Please give the confidence interval of Figure 5b and 6b.
7. The signal strength of S1 and S8 in Figure 6a does not decrease significantly as the temperature increases.
Author Response
Response to Reviewer 3 Comments
Point 1: Why choose 40 °C, 50 °C, and 60 °C for subsequent research?
Response 1: The Chinese Pharmacopoeia stipulates that the drying temperature of medicinal materials containing aromaticity components (volatile components) generally does not exceed 50 ℃. According to the results of pre-experiment, we selected these three temperatures(40°C, 50°C and 60°C.) (Nascimento et al., 2021; Vega, Fito, Andrés, & Lemus, 2007).
Point 2: Please unify the font size. For example, the font on formula three.
Response 2: Thank you for the valuable comment. We have modified the formatting in the article and marked the major changes in yellow.
Point 3: There are formatting errors in the text, such as ‘of Moslae Herba’ in the title without spaces, and the formula needs to be aligned.
Response 3: We appreciated for your valuable comment. We have fixed this error and marked the major changes in yellow.
Point 4: Please note the unit corresponding to each letter in the formula.
Response 4: Thanks to the reviewer for pointing out this issue. We have modified the unit of the letter in the formula (10) (page 5, line 165).
Point 5: What are the drying constants a, k, y, and b in Table 4?
Response 5: “a, k, y, b” are defined the model coefficients at different drying temperatures, which were related to the drying temperature. According to the model coefficients, mathematical equations can be obtained at different temperatures, which can calculate the moisture ratio at any time under specific drying conditions. (Nascimento et al., 2021; Vega et al., 2007).
Point 6: Please give the confidence interval of Figure 5b and 6b.
Response 6: We appreciated for your valuable comment. We have added confidence intervals to the Figure 5b and 6b.
Point 7: The signal strength of S1 and S8 in Figure 6a does not decrease significantly as the temperature increases.
Response 7: The signal intensity of S1 and S8 were significantly higher in the fresh samples than in the dried Moslae herba, indicating that the heating process had a significant effect on the volatile components of Moslae herba and that the signal intensity decreased slightly with increasing temperature (page14, line 401-403).
References
National Pharmacopoeia Commission. Pharmacopoeia of the People's Republic of China: Part Four [M] Beijing: China Medical Science and Technology Press, 2020
Nascimento, L. D. D., Silva, S. G., Cascaes, M. M., Costa, K. S. D., Figueiredo, P. L. B., Costa, C. M. L., . . . de Faria, L. J. G. (2021). Drying Effects on Chemical Composition and Antioxidant Activity of Lippia thymoides Essential Oil, a Natural Source of Thymol. Molecules, 26(9). doi:10.3390/molecules26092621
Vega, A., Fito, P., Andrés, A., & Lemus, R. (2007). Mathematical modeling of hot-air drying kinetics of red bell pepper (var. Lamuyo). Journal of Food Engineering, 79(4), 1460-1466. doi:10.1016/j.jfoodeng.2006.04.028

Reviewer 4 Report
The manuscript concerns the effects of the hot air thin-layer drying process on the aroma attributes of Moslae herba. I suggested that the manuscript should be submitted to Foods (MDPI). Although this manuscript contained scientific novelty, the authors were suggested to format and proofread the manuscript carefully. Too many formatting errors (fonts, equations, tables, units) were found. Which does the "final quality" in the title mean? Moreover, the authors were suggested to check the right herb name "Moslae herba" or "Herba moslae"
Author Response
Response to Reviewer 4 Comments
Point: The manuscript concerns the effects of the hot air thin-layer drying process on the aroma attributes of Moslae herba. I suggested that the manuscript should be submitted to Foods (MDPI). Although this manuscript contained scientific novelty, the authors were suggested to format and proofread the manuscript carefully. Too many formatting errors (fonts, equations, tables, units) were found. Which does the "final quality" in the title mean? Moreover, the authors were suggested to check the right herb name "Moslae herba" or "Herba moslae".
Response: Thank you very much for your offer, before that we have discussed with the editor-in-chief of Foods and he thinks that our article is suitable for the journal of molecules. We have corrected the format in the paper. The final quality refers to the overall quality of the herbal material after processing, it determines the efficacy of the herbal material. Generally speaking, the final quality of the herbal material should be consistent, only when the quality reaches unity, can the efficacy be achieved. We have checked the right herb name "Moslae herba"(Yu et al., 2020; Zhang et al., 2018).
References
Yu, W. Y., Li, L., Wu, F., Zhang, H. H., Fang, J., Zhong, Y. S., & Yu, C. H. (2020). Moslea Herba flavonoids alleviated influenza A virus-induced pulmonary endothelial barrier disruption via suppressing NOX4/NF-kappaB/MLCK pathway. J Ethnopharmacol, 253, 112641. doi:10.1016/j.jep.2020.112641
Zhang, H. H., Yu, W. Y., Li, L., Wu, F., Chen, Q., Yang, Y., & Yu, C. H. (2018). Protective effects of diketopiperazines from Moslae Herba against influenza A virus-induced pulmonary inflammation via inhibition of viral replication and platelets aggregation. J Ethnopharmacol, 215, 156-166. doi:10.1016/j.jep.2018.01.005

Reviewer 5 Report
The authors have described methods to evaluate the effects of different hot air thin-layer drying protocols on Moslae Herba, which primarily involved 3 different temperatures. They have provided insights into the optimum conditions required to obtain a high quality dried product with some loss of original flavors due to loss of volatile compounds. Using the best mathematical model, the study lays a theoretical framework of the kinetics of the drying process for Moslae Herba.
Author Response
Response to Reviewer 5 Comments
Point: The authors have described methods to evaluate the effects of different hot air thin-layer drying protocols on Moslae Herba, which primarily involved 3 different temperatures. They have provided insights into the optimum conditions required to obtain a high quality dried product with some loss of original flavors due to loss of volatile compounds. Using the best mathematical model, the study lays a theoretical framework of the kinetics of the drying process for Moslae Herba.
Response: Thank you very much for taking your valuable time to review our manuscript, we are very pleased to receive your approval.

Round 2
Reviewer 4 Report
The manuscript was formatted carefully in this version.